# Exposure to a Multilevel, Multicomponent Obesity Prevention Intervention (OPREVENT2) in Rural Native American Communities: Variability and Association with Change in Diet Quality

**DOI:** 10.3390/ijerph182212128

**Published:** 2021-11-19

**Authors:** Michelle Estradé, Ellen J. I. van Dongen, Angela C. B. Trude, Lisa Poirier, Sheila Fleischhacker, Caroline R. Wensel, Leslie C. Redmond, Marla Pardilla, Jacqueline Swartz, Margarita S. Treuth, Joel Gittelsohn

**Affiliations:** 1Department of International Health, Bloomberg School of Public Health, Johns Hopkins University, Baltimore, MD 21205, USA; lpoirie4@jhu.edu (L.P.); cwensel1@jhu.edu (C.R.W.); mpardil1@jhu.edu (M.P.); jswartz4@jhmi.edu (J.S.); jgittel1@jhu.edu (J.G.); 2Department of Pediatrics, University of Maryland School of Medicine, Baltimore, MD 21201, USA; atrude@som.umaryland.edu; 3Georgetown University Law Center, Washington, DC 20001, USA; Sef80@georgetown.edu; 4School of Allied Health, University of Alaska Anchorage, Anchorage, AK 99508, USA; lcredmond@alaska.edu; 5Department of Kinesiology, School of Pharmacy and Health Professions, University of Maryland Eastern Shore, Princess Anne, MD 21853, USA; mstreuth@umes.edu

**Keywords:** MLMC intervention, exposure, diet quality, healthy eating index, Native American

## Abstract

The OPREVENT2 obesity prevention trial was a multilevel multicomponent (MLMC) intervention implemented in rural Native American communities in the Midwest and Southwest U.S. Intervention components were delivered through local food stores, worksites, schools, community action coalitions, and by social and community media. Due to the complex nature of MLMC intervention trials, it is useful to assess participants’ exposure to each component of the intervention in order to assess impact. In this paper, we present a detailed methodology for evaluating participant exposure to MLMC intervention, and we explore how exposure to the OPREVENT2 trial impacted participant diet quality. There were no significant differences in total exposure score by age group, sex, or geographic region, but exposure to sub-components of the intervention differed significantly by age group, sex, and geographical region. Participants with the highest overall exposure scores showed significantly more improvement in diet quality from baseline to follow up compared to those who were least exposed to the intervention. Improved diet quality was also significantly positively associated with several exposure sub-components. While evaluating exposure to an entire MLMC intervention is complex and imperfect, it can provide useful insight into an intervention’s impact on key outcome measures, and it can help identify which components of the intervention were most effective.

## 1. Introduction

For over three centuries, Native Americans have faced extraordinary challenges to their health and wellbeing as a lasting result of colonization and United States (U.S.) policies that displaced them from their traditional lands and imposed westernized lifestyles. Many Native American peoples were prevented from practicing their traditional food cultures, characterized by predominantly plant-based diets supplemented with hunting, trapping, and fishing, and instead forced to adopt more sedentary agricultural practices and rely on government subsidies [1,2,3]. This transition resulted in a dietary pattern characterized by high intake of processed foods high in fat, sugar, and sodium, and low in fiber [4,5,6], followed by an exponential rise in nutrition-related chronic diseases, such as obesity and type 2 diabetes [2,7,8,9].

The resilience of Indigenous peoples in the face of such challenges has been well documented [10,11,12,13,14], including through the powerful ongoing advocacy for food sovereignty among Native American peoples [15,16]. This movement to achieve long term health and food security through nutritious, culturally adapted indigenous foods has led to the formation of many partnerships aimed at providing evidence-based nutrition interventions. Multilevel multicomponent (MLMC) community-based interventions are one such example, which has emerged as a promising approach for restoring positive health behaviors and outcomes in Native American populations [17,18,19]. A MLMC style intervention attempts to reach participants in many different settings of their lives to influence behavior by creating positive environmental and policy changes that reinforce educational components. Due to the variety of settings and stakeholders involved in a MLMC intervention, they are inherently community-based. They are also inherently complex; a challenge for MLMC interventions is that resources need to be spread over many intervention components that are delivered throughout the community. This may result in participants’ exposure to each component of the intervention being too low to achieve sufficient dose to impact outcomes [20]. Therefore, in addition to evaluating intervention effectiveness, process evaluations are important for complex MLMC interventions, as they provide insight in intervention implementation and reach of the target audience [21].

Process evaluations include measures of fidelity, reach, and dose delivered to describe the implementation of an intervention [17,21,22,23,24]. In contrast to interventions targeting a limited group of participants, community-based interventions have less control over who is exposed to the intervention and to what degree they are exposed [25]. Consequently, there can be large differences in the extent to which individuals within the community receive components of the intervention [25]. Only reporting what intervention activities or materials were delivered in the community (dose delivered and fidelity) does not necessarily represent actual individual-level exposure to the intervention. Exposure (or ‘dose received’) is a way to characterize individuals’ involvement with the intervention components, in contrast to the population-level measure of reach [25]. Assessing exposure to community-based interventions is therefore necessary to make valid claims about intervention effectiveness and impact.

Previous work in indigenous communities, especially in food stores, indicates that intervention exposure is positively associated with improved health outcomes, health behavior, and/or psychosocial outcomes [26,27,28,29,30]. For example, in the Navajo Healthy Stores program, exposure to separate intervention components as well as an overall exposure score were reported, with highest mean exposures to the intervention logo, posters, and educational displays [26]. Overall exposure scores were positively associated with changes in healthy food purchasing and intentions to consume healthy foods, and negatively associated with change in participants’ Body Mass Index (BMI). Although numerous community-based diet interventions have reported on exposure, most have only done so for one type of institution (i.e., food stores); thus, little is known about individual-level exposure to multiple interventions delivered simultaneously in different settings in a community. 

The OPREVENT2 obesity prevention intervention was implemented through local food stores, worksites, schools, community action coalitions, and by social and community media. A report of the process evaluation of the OPREVENT2 intervention provided insight about fidelity, reach and dose delivered [31]; however, it did not assess individual-level exposure to the intervention (dose received). In the present study, we provide a methodology for calculating and combining component-level exposure scores into a single intervention-level exposure score that can be used to contextualize the impact results of the OPREVENT2 intervention. Since a primary focus of OPREVENT2 was on improving dietary quality [32], we examined whether exposure was associated with changes in Healthy Eating Index (HEI-2015). We hypothesized that participants with higher exposure to the intervention would show significant improvement in HEI scores. 

The purpose of this paper is therefore to: Develop an aggregated exposure score for the multilevel multicomponent OPREVENT2 intervention;Assess exposure to the OPREVENT2 intervention and explore variability in intervention component exposure based on participant characteristics/demographic factors; and,Examine the association between intervention exposure and changes in diet quality (HEI-2015).

## 2. Methods

### 2.1. Study Design

OPREVENT2 was a MLMC community randomized obesity prevention trial carried out in six rural Native American communities from 2016 to 2019. All communities were located on Indian reservation– two in the Midwest and four in the Southwest. The study was approved in 2016 by the Johns Hopkins School of Public Health Institutional Review Board (IRB) (IRB# 00007028), the Indian Health Service IRB (IRB# NI6-N-04), and the Navajo Nation Human Research Review Board (IRB# NNR-16-245). 

### 2.2. Community Recruitment

Prior to obtaining the grant to fund OPREVENT2, all tribes in the two regions were invited, with 12 in the Southwest and 6 in the Midwest responding in favor. Out of 18 total interested, 6 tribes followed through with a resolution and memorandum of understanding to participate. Collaborative partnerships were established with each tribal community and approval of the proposed research was obtained. After baseline data collection, three communities were randomized to receive the intervention immediately, while comparison communities received the same intervention after follow-up data collection.

### 2.3. Participants and Recruitment

Approximately 100 participants from each of the six communities were recruited for the evaluation sample (*n* = 601). To meet inclusion criteria, participants had to be the main food shopper or food preparer for their household, 18–75 years old, not pregnant, and have no plans to move away from the community for at least two years. 

All interventionists and data collectors participated in a weeklong in-person training prior to starting the baseline data collection and, prior to follow-up data collection, attended a 2-day booster training. These trainings emphasized methods for consistent and accurate recording of all data, as well as uniform delivery of intervention components across study sites. The interventionists and data collectors were tribal members familiar with the language, culture and customs in each study site. As speakers of the local tribal languages, they were able to translate during interactions and interviews if requested by a participant. The consent form was read to the participant and a written informed consent was obtained for all participants prior to baseline and follow-up interviews. 

### 2.4. Intervention

The OPREVENT2 intervention was implemented over the course of 18 months, in six phases each lasting 2–4 months. The different phases were used to focus on specific educational messages or activities, such as choosing lean proteins and high-fiber foods, or encouraging participation in a workplace walking group. The intervention’s multiple components were designed to reinforce one another and reach people through multiple facets of their daily life and activities. At least one worksite in each community hosted physical activity opportunities, displayed educational posters about healthy diets, and distributed informational booklets on diet and physical activity. Interventionists worked with local food store managers to ensure that healthy food options were available and that labels were hung to highlight the healthier choices. A school curriculum was delivered in grades 2–6, with children envisioned as motivators of their adult relatives, and therefore able to act as agents of behavior change within their families. Key intervention messages were reinforced through postal newsletters, radio announcements, social media posts, and through the formation of community action coalitions. Further details about the intervention design [32] and implementation [31] have been described elsewhere.

### 2.5. Measurements

Data on sociodemographic variables were collected at baseline, using an Adult Impact Questionnaire. Variables included age (continuous), sex (male/female), current smoking status (yes/no), employment status (full time/part time/seasonal/unemployed/student/retired/disabled), education level (less than high school/diploma or GRE/some college/undergraduate degree/graduate degree), and household size (continuous). A Material Style of Life (MSL) score was calculated as a proxy for socioeconomic status. The MSL score was a summative total of 18 questions about material possessions of the respondent’s household, for example, “How many working televisions are in your home?” and “How many working computers/laptops/tablets are in your home?”. MSL scores ranged from 1 to 45 (mean 19, SD 8), with a higher MSL score indicating more material possessions in the participant’s household.

Dietary intake was measured using a 113-item semi-quantitative Block Food Frequency Questionnaire (FFQ) that probed for foods consumed in the past 30 days at baseline and follow up. The FFQ was adapted from one used in the Strong Heart Study [33] and modified during the formative phase of OPREVENT2 to include foods that were culturally relevant in the study communities, such as piñon nuts in the Southwest and venison in the Midwest. Completed FFQs were sent to Nutrition Quest (Berkeley, CA, USA) for processing and analysis of nutrient and food group intakes. We then used these data to calculate a Healthy Eating Index Score (HEI-2015), based on how closely reported intake aligns with the United States Department of Agriculture (USDA) 2015–2020 Dietary Guidelines for Americans (DGA) on a scale of 1–100 [34]. The HEI-2015 calculation methods and evaluation in OPREVENT2 participants at baseline have been described in detail in a previous study [35].

Data on exposure was obtained at post-intervention data collection for the intervention and comparison groups, using the Intervention Exposure Evaluation Instrument (IEEI). An overview of intervention components, type of data collected and scoring is presented in Table 1. The IEEI included questions to assess to what extent respondents reported their own exposure to each component of the intervention: Shelf labels (five questions), posters (20 questions), radio announcements (1 question), social media (6 questions), store visits (1 question), schools (1 question), handouts (21 questions), booklets (7 questions), newsletters (3 questions), educational displays (5 questions), giveaways (12 questions), taste tests (8 questions), and worksite (1 question). For educational materials (shelf labels, posters, educational displays, handouts, booklets, newsletters) respondents were asked whether they had seen/read/received the material. Respondents were shown images of the selected educational materials during questionnaire completion to facilitate recall. For exposure to the food store component, respondents were also asked how many times they visited each of the stores that participated in the intervention in the last 30 days. For worksites, respondents were asked to indicate whether they had worked at one or more of the worksites in which the intervention was delivered. For the school component, respondents indicated whether they had a child that went to one of the elementary schools participating in the intervention. In addition, five dummy questions (often referred to as “red herring” questions), related to educational materials not used in OPREVENT2, were added to the questionnaire to address response bias and improve validity of the responses. 

#### 2.5.1. Generating Component-Level Exposure Scores

An exposure score was created for each intervention component (Table 1). To make the components comparable, for each component we divided the exposure into bivariates, tertiles or quartiles, depending on the question. For example, the IEEI listed nine possible taste tests that respondents could have sampled. Based on distribution of the responses we divided the scores into tertiles: a score of 0 for respondents that did not participate in any taste tests, a score of 1 for respondents that participated in between one and three taste tests, and a score of 2 for respondents who participated in more than 3 taste tests (Table 1). For the worksite component we used two categories, where people either worked at one or more of the worksites in the past 12 months (score 1) or did not (score 0). As each component has different possible scores, and we wanted to make the components comparable with each other, we re-scaled all component exposure scores so each component was scored between 0 and 1. These re-scaled component exposure scores were used to assess levels of exposure to the different intervention components. 

#### 2.5.2. Calculating an Intervention-Level Exposure Score

To assess overall exposure to the OPREVENT2 intervention, we summarized all intervention components into a total exposure score. We used a weighting scheme to assign more relative weight to different intervention components, based on the type of component/activity being more or less interactive. Based on the underlying intervention theory and discussions with the OPREVENT2 interventionists and researchers on intervention delivery, we placed all components along a continuum that ranged from more interactive to less interactive. Since the level of interactiveness was specific to the OPREVENT2 intervention, the ‘ranking’ of each component required several rounds of in-depth discussion and deliberation with the interventionists who had delivered the intervention. For instance, social media may have been less interactive than newsletters due to the remote nature of the communities and lack of cell phone and internet service. More interactive intervention components, in which participants spent a longer time engaging with the component, were thought to be more important for bringing about changes in participants’ outcomes than more passive, solely knowledge-transfer components. Therefore, we weighted intervention components on the right side of the spectrum (more interactive) higher than components on the left side of the spectrum (more passive) (Figure 1). For instance, worksites were a convenient place to implement many intervention activities and interact with participants, because participants spent many hours there; therefore, worksites were placed on the right side of the spectrum. Conversely, posters, passively displayed in the community and used to convey the main intervention messages, were placed on the left side of the spectrum. To generate the overall intervention-level exposure score, we multiplied the re-scaled component exposure scores with their specific weighting factor (either 1, 2, 3 or 4) and subsequently we summed all weighted component scores. The total exposure score ranged from 0 to 28, with a higher score indicating greater exposure to the intervention. 

### 2.6. Statistical Analysis

After reviewing the exposure data for the entire study, it was clear that none of the participants from comparison communities reported exposure to any components of the intervention. Since the main objective of this paper is to examine exposure and its correlates, we decided to conduct the exposure analysis utilizing only data from participants who completed follow-up interviews in intervention communities (*n* = 243). The retention rate from baseline to follow up was 81%. Participants were excluded from the analysis if they reported seeing ≥80% of the red herring questions (*n* = 7) or had missing exposure data (*n* = 2), leaving a final sample of *n* = 234. One-way ANOVA was used to compare mean scores for each exposure component by age group, sex, and geographic region. 

The HEI-2015 analysis was conducted on the same sample as the exposure analysis (*n* = 234), but those reporting intakes outside the range of 500–7000 kCal per day (*n* = 7) were excluded from the dataset, yielding a final sample size of *n* = 227. Each participant’s baseline HEI-2015 score was subtracted from their follow-up HEI-2015 to calculate a HEI-2015 change score. The distribution of residuals was checked and met assumptions for normality, then linear regression models were used to regress change in HEI-2015 on each of the exposure sub-components and total exposure score, controlling for age, sex, smoking status, baseline HEI-2015 score, community, and MSL score.

## 3. Results

### 3.1. Exposure to the OPREVENT2 Intervention 

Overall exposure to the OPREVENT2 intervention was moderate (mean 11.66 ± 6.71 on a scale of 0–28), with store visits, posters, and handouts being the components to which participants reported the highest exposure, and schools and social media the components to which participants in the evaluation sample were least exposed. 

There were no significant differences in total exposure score by age group, sex, or geographic region, but there were notable demographic differences in several exposure component scores. Participants in the oldest age group (≥60 y) reported significantly lower exposure to the school component of the intervention than those in either of the other age groups. Females reported significantly higher exposure to newsletters than men. Additionally, compared to those in the Midwest, participants in the Southwest reported significantly higher exposure to taste tests, school curriculum, handouts, booklets, and radio announcements, but significantly lower exposure to the worksites and food stores (Table 2).

### 3.2. Association between Exposure and Changes in Diet Quality

Participants with the highest overall exposure scores had an average of 3.61 points (*p* = 0.049) higher change in HEI-2015 from baseline to follow up compared to those who were least exposed to the intervention. Improved HEI-2015 scores were also positively associated with exposure to four of the 13 intervention exposure sub-components. (Table 3). A higher exposure to educational displays, handouts, posters, and radio announcements was associated with a significant improvement in HEI-2015 score. There were no significant changes in HEI-2015 scores among participants in the three comparison communities that did not receive the intervention.

## 4. Discussion

This is one of the first studies to describe a method for generating an aggregated exposure score for a MLMC intervention. Mean level of exposure varied greatly across intervention components, with participants reporting highest exposure to food stores and posters. This finding makes sense given that the food store component was implemented with high reach, dose delivery and fidelity throughout the intervention [31], and that posters were hung in many locations throughout the community, remaining on display for extended periods of time. In contrast, the components to which participants reported least exposure—schools and social media—were found to have been implemented with moderate-to-high dose delivered, while reach and fidelity were moderate (schools) or low (social media) [31].

Exposure to several components differed significantly by age, sex, or geographic region. This is expected, since not all people interact in the same ways with institutions in their communities. Indeed, a strength of MLMC interventions is that they can reach a broad range of people and demographic groups in differing contexts, with some components more targeted than others [36]. 

Exposure to the OPREVENT2 intervention was associated with a significant positive change in HEI-2015 scores. Those most exposed gained an average of 3.61 HEI points from baseline to follow-up, which can be interpreted as a meaningful impact on diet quality [37]. Most of the intervention components that were associated with a significant change in HEI-2015 scores were components delivered in food stores, including educational displays, posters, and handouts, which suggests that print materials and point-of purchase interventions might be a promising way to impact diet quality in rural Native American communities.

This study has several limitations. The use of FFQs to evaluate diet quality is imperfect and subject to recall bias and social desirability bias [38]. Measurement and development of a complex exposure measure is also subject to recall bias since we relied on participants’ self-reported memory of being exposed to each component, which may lead to an underestimation of actual exposure. Adult exposure to the school component may have been underestimated because we only asked whether each participant had a child in one of the intervention schools, which does not account for children’s interactions with other extended family members. Furthermore, if participants indicated they saw or remembered intervention components, we assumed that they fully absorbed and interacted with the intervention, which might not be true. While the interventionists, who were intimately familiar with the study communities, gave input on exposure score weighting and interactiveness of each component, time and resources did not allow for direct consultation with community members during exposure score development. Finally, the sample size, when divided into demographic subgroups for comparison, may not have been adequate to detect significant differences in exposure (Appendix A). 

These findings should not be interpreted as generalizable to all rural Native American communities, as there are 574 federally recognized American Indian/Alaska Native tribes in the USA, representing a vast diversity of languages, cultures, and traditions. Acknowledging this diversity may help to explain and contextualize some of the significant regional differences observed in exposure to the OPREVENT2 intervention. 

## 5. Conclusions

The OPREVENT2 intervention was successful in improving participants’ diet quality, and those with highest exposure to the intervention had the greatest improvement. Future intervention efforts should evaluate the balance between resources required for implementation and achieved exposure. For example, the school curriculum was a time- and resource-intensive component of OPREVENT2 that achieved very little exposure and therefore may not be worth including in future interventions.

While evaluating exposure to an entire MLMC intervention is complex and imperfect, it can provide useful insight into an intervention’s impact on key outcome measures, and it can help identify which components of the intervention were most effective. This work can be seen as one important piece of the larger ongoing effort in Native American communities to prioritize strengths and resources for interventions with the highest potential to support and restore the health of their people.

## Figures and Tables

**Figure 1 ijerph-18-12128-f001:**
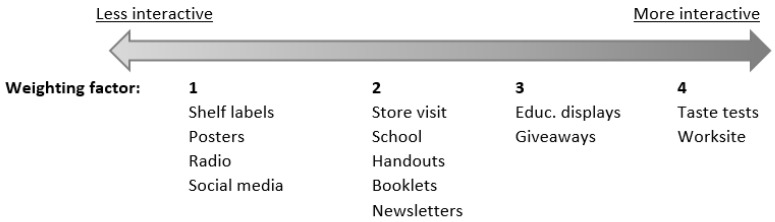
Continuum of interaction/engagement within intervention components, with corresponding factors (one to four) used in the weighting scheme.

**Table 1 ijerph-18-12128-t001:** Exposure scores for OPREVENT2 intervention materials and activities.

Intervention Component	Intervention Material/Activity	Exposure Data Collected	Coding of Exposure (before Re-Scaling)
**Shelf Labels**	Labels put up in food stores to promote healthful foods and beverages linked to intervention phase	Number of shelf labels seen	None = 0 1–2 shelf labels = 1 3–5 shelf labels = 2
**Posters**	Posters put up at multiple places in the community (e.g., laundromats, community centers)	Number of posters seen and/or read	None = 0 1–8 posters = 1 9–20 posters = 2
**Radio Announcements**	Announcements broadcasted at community radio station	Heard any radio announcements on local radio stations (y/n)	No = 0 Yes = 1 or 2 ^a^
**Social Media**	Messages related to intervention phase posted on Facebook, Instagram and Twitter	Followed or seen OPREVENT2 on Facebook, Instagram and/or Twitter (y/n)	No = 0 Yes = 1 ^b^
**Food Store Visit**	Stocking of healthful foods and beverages, educational materials, promotional activities at point of purchase	Number of times each store was visited in last 30 days ^c^	None = 0 1–9 store visits = 1 10–20 visits = 2 >20 visits = 3
**School**	Elementary school curriculum for grades 2–6	Have a child who attended a school (y/n)	No = 0 Yes = 1 or 2 ^d^
**Handouts**	Educational leaflet handed out by interventionists at multiple places in the community (e.g., laundromats, community centers)	Number of handouts received and/or read	None = 0 1–7 handouts = 1 8–21 handouts = 2
**Booklets**	Booklets handed out by interventionists at different places in the community	Number of booklets received and/or read	None = 0 1–4 booklets = 1 5–7 booklets = 2
**Newsletters**	Newsletter send to the evaluation sample’s home	Number of newsletters seen and/or read	None = 0 1–2 newsletters = 1 3 newsletters = 2
**Educational Displays**	Interactive displays including posters, set up at food stores and worksites	Number of educational displays seen and/or read	None = 0 1–2 educational displays = 1 3–5 educational displays = 2
**Giveaways**	Small gifts handed out by interventionists at multiple places in the community (e.g., at food stores and worksites)	Number of giveaways received ^e^	None = 0 1–5 points = 1 6–10 points = 2 >10 points = 3
**Taste tests**	Taste tests provided by interventionists at multiple places in the community (e.g., at food stores and worksites)	Number of taste tests participated in/seen	None = 0 1–3 taste tests = 1 4–9 taste tests = 2
**Worksite**	Coffee station makeover, educational materials, interventionist visits, pedometer challenge	Worked in one or more of the worksites in past 12 months (y/n) ^f^	Not working at worksite = 0 ^g^ Working at ≥1 worksite = 1

^a^ Score depends on extent to which radio announcements were broadcasted. Radio in community that stopped broadcasting the announcements halfway through intervention = 1, radio in community that broadcasted throughout intervention period = 2. ^b^ Respondent followed at least one of the social media accounts and/or saw at least one of the example social media posts. ^c^ For two stores, the number of visits were multiplied by 1 as they did not implement all components of the intervention. For the other stores, the number of visits was multiplied by 2, as they implemented majority of intervention components. The outcomes were added to create the food store exposure score. ^d^ Score depends on extent to which school implemented OPREVENT2 curriculum. Not implemented = 0, partly implemented = 1, majority/fully implemented = 2. ^e^ Giveaways were assigned different scores based on the amount of interaction needed to receive the giveaway (tiers), with score 1 as lowest interaction and score 3 as highest interaction. These scores are added to create the giveaway exposure score. ^f^ Answer choices were yes or no. ^g^ One worksite was assigned a score of 0 as the intervention was not implemented in this worksite.

**Table 2 ijerph-18-12128-t002:** Re-scaled component exposure scores (Mean (SD)) for 13 intervention components and the Total Exposure Score (Mean (SD), not re-scaled, range 0–28) to the OPREVENT2 intervention for the total intervention group, and by age, sex, and region.

Exposure Component	Total Intervention Group (*n* = 234)	Age Group			Sex		Region	
		18–35 (*n* = 73)	36–59 (*n* = 123)	60+ (*n* = 38)	Male (*n* = 63)	Female (*n* = 171)	South-West (*n* = 162)	Midwest (*n* = 72)
Shelf Labels	0.41 (0.41)	0.43 (0.41)	0.39 (0.39)	0.42 (0.46)	0.34 (0.41)	0.44 (0.40)	0.42 (0.41)	0.39 (0.41)
Posters	0.59 (0.37)	0.57 (0.38)	0.60 (0.38)	0.62 (0.34)	0.59 (039)	0.60 (0.37)	0.62 (0.38)	0.53 (0.36)
Radio Announcements	0.38 (0.48)	0.31 (0.46)	0.43 (0.48)	0.38 (0.49)	0.33 (0.46)	0.41 (0.48)	**0.53** **(0.50)** ^a^	**0.06** **(0.17)** ^a^
Social Media	0.16 (0.36)	0.17 (0.38)	0.16 (0.37)	0.11 (0.31)	0.11 (0.32)	0.17 (0.38)	0.17 (0.37)	0.14 (0.35)
Food Store Visits	0.67 (0.29)	0.62 (0.29)	0.69 (0.29)	0.69 (0.26)	0.65 (0.28)	0.68 (0.29)	**0.63** **(0.27)** ^b^	**0.74** **(0.31)** ^b^
School	0.12 (0.28)	**0.18** **(0.31)** ^c^	**0.13** **(0.28)** ^d^	**0.00** **(0.00)** ^c,d^	0.08 (0.24)	0.14 (0.29)	**0.17** **(0.32)** ^e^	**0.01** **(0.08)** ^e^
Handouts	0.50 (0.40)	0.43 (0.41)	0.52 (0.40)	0.57 (0.41)	0.42 (0.40)	0.53 (0.40)	**0.54** **(0.40)** ^f^	**0.42** **(0.41)** ^f^
Booklets	0.40 (0.40)	0.36 (0.39)	0.41 (0.40)	0.43 (0.41)	0.35 (0.40)	0.42 (0.40)	**0.45** **(0.40)** ^g^	**0.29** **(0.36)** ^g^
Newsletters	0.42 (0.44)	0.40 (0.46)	0.40 (0.42)	0.53 (0.45)	**0.33** **(0.40)** ^h^	**0.46** **(0.44)** ^h^	0.41 (0.43)	0.45 (0.46)
Educational Displays	0.34 (0.39)	0.30 (0.38)	0.37 (0.40)	0.34 (0.37)	0.30 (0.35)	0.36 (0.40)	0.37 (0.39)	0.28 (0.37)
Giveaways	0.43 (0.39)	0.36 (0.37)	0.46 (0.39)	0.47 (0.40)	0.39 (0.39)	0.45 (0.38)	0.42 (0.38)	0.46 (0.41)
Taste Test	0.43 (0.41)	0.35 (0.39)	0.48 (0.42)	0.45 (0.42)	0.43 (0.43)	0.43 (0.41)	**0.49** **(0.41)** ^i^	**0.30** **(0.40)** ^i^
Worksite	0.46 (0.50)	0.45 (0.50)	0.49 (0.50)	0.39 (0.50)	0.41 (0.50)	0.48 (0.50)	**0.39** **(0.49)** ^j^	**0.62** **(0.49)** ^j^
Total Exposure Score	11.66 (6.71)	10.56 (6.56)	12.28 (6.72)	11.78 (6.91)	10.44 (6.82)	12.11 (6.64)	12.09 (6.66)	10.70 (6.78)

Differences tested with ANOVA indicated in bold: ^a,e^
*p*-diff < 0.0001; ^b^
*p*-diff = 0.008; ^c^
*p*-diff = 0.003; ^d^
*p*-diff = 0.034; ^f^
*p*-diff = 0.048; ^g^
*p*-diff = 0.006; ^h^
*p*-diff = 0.041; ^i,j^
*p*-diff = 0.001.

**Table 3 ijerph-18-12128-t003:** Associations between exposure to OPREVENT2 and change in diet quality (Healthy Eating Index 2015 score, HEI-2015) (*n* = 227) *.

Exposure Component ^	Change in Total HEI-2015 (Points) Post-Intervention-Baseline		
	β	SE	*p*-Value
Shelf Labels	0.34	1.19	0.772
Posters	3.70	1.27	0.004
Radio	3.10	1.07	0.004
Social Media	2.31	1.34	0.085
Food Store Visits	0.01	1.73	0.999
Schools	3.39	1.82	0.064
Handouts	2.50	1.20	0.039
Booklets	1.96	1.24	0.116
Newsletters	0.82	1.10	0.456
Educational Displays	3.39	1.21	0.006
Giveaways	−0.12	1.30	0.924
Taste Test	1.18	1.18	0.316
Worksite	1.13	0.99	0.253
Total Exposure Score (re-scaled)	3.61	1.90	0.049

* Associations are adjusted for baseline HEI score, sex, age, MSL score, smoking, and community. ^ Re-scaled component exposure scores and total exposure score (0–1).

## Data Availability

The data are not publicly available.

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
