# Peer review of "Exposure to a Multilevel, Multicomponent Obesity Prevention Intervention (OPREVENT2) in Rural Native American Communities: Variability and Association with Change in Diet Quality"

_ijerph, 2021, doi:10.3390/ijerph182212128_

Round 1
Reviewer 1 Report
The paper examined how well the various components of an intervention trial were implemented in terms of if the participants were exposed to the material, and how strongly each component impacted the overall goal of improved health. It is an interesting concept and the results could prove quite useful in the design of future studies. However, it is fairly limited in its analyses. 'Exposure' is treated fairly broadly, for example, having seen a poster is not treated differently from having read it. Additionally, only three types of demographic information were taken into account. More information about the participants could have strengthened the weight of the conclusions, for instance 'schools' may have had such a low exposure score because very few of the participants had school aged children; it is not possible to determine if this is the case with the information given, so stating that this was a misuse of time and resources is unfounded. While the basis for the paper is promising, it is not investigated as thoroughly as perhaps it could be.
I do not understand why the name of the trial was concealed in the manuscript. It is clearly stated as OPREVENT2 intervention in the article information and cited references.
[Lines 23-25] Were there any participant characteristics considered other than age, sex and geographic region?
[Lines 25-28] Is the correlation between exposure score/sub-components and diet quality statistically significant?
[Abstract] Are there particular exposure sub-components that are more significant that may be worth naming in the abstract?
[Lines 18, 36 and 336] Please be consistent with acronyms.
[Tables 1 and 2] The footnotes should be labelled in order of appearance
[Figure 1] Were the participants themselves consulted when the level of engagement of each component was determined?
[Lines 119-122 and 254-256] This is unclear. If the comparison communities did indeed only receive intervention after the data was collected, meaning they had not yet had the opportunity to be exposed to the various intervention strategies, of course their scores would all be zeros. Why is this a surprising fact? These comparison communities would be a control group for the overall intervention, but their relevance to investigating the effectiveness of the various components of the intervention, the point of this paper, is unclear. Why could their data not be collected and added to this study after their intervention had taken place?
[Lines 279-281] Older (>60 years of age) participants not having primary school aged children is hardly surprising. Was the question adjusted for inclusion of grandchildren?
[Line 286] If the total intervention score was not rescaled, wouldn't the range be to 56, rather than 28?
[Lines 293-296] These two sentences repeat the same information.
[Tables 1-3] Be consistent with footnote symbols.
[Table 3] What does beta refer to? Why is SE used here when SD was used in Table 2?
[Supplementary Table S1] Why was geographic region not included?
Reviewer 2 Report
I’m a little unclear on whether or not you compared diet quality (HEI) of the intervention group to the control group or comparison group. I can’t find anything in the results to suggest that you did and. I understand that it was clear that the control group was not exposed to the intervention, but that’s the point of having a control group, yes? Unless I’m missing something, I think the lack of comparing HEI to the control group is a major flaw and this manuscript shouldn’t be published until those data are presented.
It is clear that HEI was linked to some exposure components but not others. Do you think that some of the components could be eliminated and if so, which ones? I think that information would be useful to others who are doing similar interventions in similar communities.
Overall, I think the content is somewhat non-significant to science. The generation of an aggregated exposure score is not a new concept to multiple fields of science as well as the field of program evaluation. Furthermore, I question the methodology of weighting the various components of exposure based on how the interventionists view their relative level of exposure rather than conducting a focus group with actual participants/recipients of the intervention.
Reviewer 3 Report
Overall, this is a very interesting and important topic addressing health promotion in Native American community by focusing on multilevel multicomponent intervention. Introduction, methods and results and discussion section are concise and offer great clarity and depth.
There are no major revisions required.
Minor comment: It is not surprising that females comprised on about 75% of the sample in this intervention. What are authors’ thoughts on including more men in future interventions? Additionally, since the participant had to be primary food shopper or preparer, would the effects of intervention on participants’ food shopping and preparing behavior translate to health behaviors of family members?
Reviewer 4 Report
This paper demonstrates that direct personal contact is superior to passive impersonal means of communication and that effective social change requires massive inputs of human time and effort; this could be addressed when placing this study in a "real-world" operational context.
